# Exosomes as a Potential Tool for Supporting Canine Oocyte Development

**DOI:** 10.3390/ani10111971

**Published:** 2020-10-27

**Authors:** Seok Hee Lee, Islam M. Saadeldin

**Affiliations:** 1Center for Reproductive Sciences, Department of Obstetrics and Gynecology, University of California San Francisco, San Francisco, CA 94143, USA; 2Department of Animal Production, College of Food and Agriculture Sciences, King Saud University, Riyadh 44511, Saudi Arabia; isaadeldin@ksu.edu.sa; 3Department of Comparative Medicine, King Faisal Specialist Hospital & Research Centre, Riyadh 11211, Saudi Arabia

**Keywords:** oviduct, dog, exosomes, extracellular vesicles, oocytes development

## Abstract

**Simple Summary:**

To date, extracellular vesicles, including exosomes, have markedly gained attention in scientific research because of their physiological homogeneity as well as stability for transporting regulatory molecules to recipient cells. Recently, it has been shown that exosomes impact gametes and embryo development in several mammalian species; however, there is still scant information on the physiological effects of exosomes on the canine reproduction system. In this regard, we elucidate the possible roles of exosomes involvement in oviduct and cumulus-oocyte complexes mutual communications and how oviduct regulates their development via molecular signaling pathways.

**Abstract:**

The canine oviduct is a unique reproductive organ where the ovulated immature oocytes complete their maturation, while the other mammals ovulate matured gametes. Due to their peculiar reproductive characteristics, the in vitro maturation of dog oocytes is still not wellestablished compared with other mammals. Investigations of the microenvironment conditions in the oviductal canal are required to establish a reliable in vitro maturation system in the dog. Previous studies have suggested that the oviduct and its derivatives play a key role in improving fertilization as well as embryo development. In particular, the biological function of oviduct-derived exosomes on sperm and early embryo development has been investigated in porcine, bovine, and murine species. However, the information about their functions on canine cumulus-oocyte complexes is still elusive. Recent canine reproductive studies demonstrated how oviduct-derived extracellular vesicles such as microvesicles and exosomes interact with oocyte-cumulus complexes and how they can play roles in regulating canine cumulus/oocyte communications. In this review, we summarize the physiological characteristics of canine oviduct-derived exosomes and their potential effects on cumulus cells development as well as oocyte in vitro maturation via molecular signaling pathways.

## 1. Exosomes as a Mediator of Cell-to-Cell Communication

Since the extracellular vesicles (EVs) were first discovered 50 years ago in plasma [1], it has been demonstrated that all biological fluids possess such membranous nanovesicles that play key roles in cellular functions and mediate intercellular communication [2,3,4]. The typical EVs contains protein, lipids, and genetic materials enclosed by a lipid bilayer membrane that originated from the plasma membrane [5]. The EVs content are determined by several factors such as the type, physiological, and pathological status of the donor cells [6]. Proteins are an important part of EVs since the inner and outer domains of proteins determine the functional capacity of EVs [7]. Some proteins also determine the EVs origin and target, including antigen presentation proteins, immunoglobulins, cell surface proteins, and cell-to-cell specific signaling proteins [8,9]. Apart from proteins, many lipids are found in EVs; however, still, their composition has not been studied compared with the protein constituent. EVs contain high contents of cholesterol, ceramide, or other sphingolipids, and phosphoglycerides with long and saturated fatty-acyl chains [10]. The lipid content contributes to the biogenesis of the EVs by transporting hormone-like molecules and regulating homeostasis in the recipient cells [11]. In addition, it has been demonstrated that EVs contain a large amount of RNAs such as mRNA, microRNAs (miRNAs), and long non-coding RNA [12]. In particular, miRNAs play an important role in regulating gene expression in recipient cells [13]. Moreover, a recent study proved that double-stranded DNA has been found in the EVs derived from the cancer cells, and the release of DNA also maintains the homeostasis in recipient cells [14]. The polysaccharides and glycans are located on the outer surface of EVs, as the least components of EVs. These factors are composed of mannose, α-2-3 and α-2-6 sialic acids, complex-n-linked glycans, and polylactosamine [15]. Among the many types of EVs classified according to their size and origin, exosomes are the smallest in size (30–100 nm indiameter) and they can serve as vehicles for the transfer of selectively sorted cellular substances and molecular cargos such as mRNA, miRNAs, DNA, and lipids to the recipient cells [2,3,4,16,17]. It is wellknown that these molecules transported by exosomes can mediate specific physiological functions and pathways in cells. In particular, exosomes mediate juxtacrine, paracrine, and endocrine communication for cell development, maintenance, and regeneration [18,19]. In addition, a recent study demonstrated that exosomes can be used for therapeutic strategies because of their remarkable physiological potential with low-cost analysis [20]. Therefore, due to the pleiotropic and bio-functional roles of exosomes in numerous cellular systems, interest in exosomes as potential biomarkers has exponentially increased, including for the reproductive system.

## 2. The Potential Role of Exosomes in the Field of Reproduction Biology

### 2.1. Folliculogenesis and Oogenesis

The multiple steps of physiological communication among developing oocytes, surrounding follicular cells consisting of cumulus granulosa, mural granulosa and theca cells, and follicular fluid have been considered as a key process for successful oocyte development. The oocyte resumes meiotic resumption during folliculogenesis and only the dominant follicle can produce the mature oocyte [21,22]. There is a constant exchange of molecular signals between the oocytes and the somatic cells during oocyte maturation, and this communication is mediated by gapjunctions established between follicular somatic cells, and between the oocyte and cumulus cells [23]. The infiltration of several capillaries provides several important plasma components to form follicular fluid, fundamental to oocyte development and to the somatic follicular cells and oocyte [24]. The follicular fluid consists of various types of proteins, steroid hormones, enzymes, growth factors, and cytokines that regulate intercellular communication between the oocyte and follicular somatic cells [25,26]. 

To date, numerous studies have identified the EVs in follicular fluids. In 2012, for the first time, it was demonstrated that follicular fluid had exosomes/microvesicles containing miRNAs and proteins in the horse [27], and more recently, they proved the role of hormonal-regulated miRNAs within different ovarian follicular cells as well as follicular fluid, which suggesting a cell-to-cell communication within the ovarian follicles [28]. Specifically, miRNAs are critically involved in RNA silencing and post-transcriptional regulation of gene expression in the cells [29,30]. It has been demonstrated that the changes in miRNAs expression were positively correlated with progesterone synthesis in bovine follicular fluids; however, it was associated with lower mitochondrial function in the oocytes [31]. Moreover, the same authors showed that miRNAs regulated the maternal mRNA storage and synthesis in the bovine ooplasm [31]. Interestingly, specific miRNAs were found to be essential for follicular growth and oocyte developmental competence; miR-769, miR-1343, miR-450a, miR-204, miR-1271, and miR-451 were found in bovine follicular fluid and regulate follicular growth and acquisition of oocyte developmental competence [31]. Likewise, deep RNA sequencing revealed several miRNAs involved in follicular growth and oocyte developmental competence in porcine such as MiR-205, miR-16, miR-148a-3p, and miR-125b [32] and in bovine such as miR-654-5p, miR-640, miR-19b-1, and miR-29c [33]. In humans, miR-214, miR-454 and miR-888 differential expression were associated with good quality embryo and successful IVF cycles [34,35,36].

Moreover, the uptake of EVs by bovine granulosa cells and their presence in the cytoplasm of cumulus cells/oocytes have been observed, indicating the biological functions of follicular fluid EVs as a signaling mediator [37] (Figure 1). According to this work, a total of 280 miRNAs were identified in the EVs, and some of them were involved in IGF2 signaling pathway, which is related to folliculogenesis and steroidogenesis. Moreover, another recent study identified the exosomal fraction from bovine follicular fluid, and particularly, 25 miRNAs were differentially expressed in exosomes and were involved in the signaling pathways related to follicular development and oocyte growth [33]. In human, 37 upregulated exosomal miRNAs were identified in the follicular fluid and they were involved in follicular maturation [36]. Recently, it has been suggested that culturing cumulus-oocyte complexes with follicular fluid exosomes significantly ameliorated the negative effects of heat shock on cleavage and blastocyst development [38]. Furthermore, bovine follicular fluid EVs promoted cumulus-oocyte complex expansion through regulating the expression of *PTGS2*, *PTX3*, and *TNFAIP6* [39]. Likewise, in the last few years, numerous research studies have demonstrated that follicular fluid EVs are involved in the regulation of multiple pathways controlling folliculogenesis and oocyte meiotic resumption in an endocrine-dependent manner. To date, as most of the EVs studies have been performed using animal models, a series of miRNAs and proteins has been identified specifically, which represent an essential resource in human reproductive research. As described in Table 1, recent studies have suggested that the overall reproductive processes such as folliculogenesis, gametogenesis, and embryogenesis are potentially regulated through EVs such as exosomes and microvesicles. Despite all these achievements, our knowledge about biological communication mediated by exosomes/microvesicles within the ovarian follicles still remains elusive. It is wellknown that follicular fluid contains numerous factors produced by the oocyte and surrounding somatic cells. Therefore, a comprehensive characterization of follicular fluid derived exosomes would be a valuable target to clarify some unexplored aspects of biological communications controlling the critical steps during oocyte development and the entire ovarian physiology.

### 2.2. Embryogenesis

The oviduct is the place where fertilization occurs and embryo development takes place in mammalian species [40]. Importantly, the major embryonic gene activation occurs during the early embryo development through their passage in the oviduct [41]. The embryos start to transcribe actively and their dependency on the maternal mRNA decreases during this phase. Therefore, the oviduct has been considered as the most suitable microenvironment for early embryo development, and numerous research studies have unveiled its dynamic functions [42]. The oviductal epithelium is composed of ciliated and secretory cells that are responsible for secreting different components of oviductal fluid such as proteins and metabolic substrates including pyruvate, glucose, and amino acids [43,44], and significant gains in our understanding of their interactions with the embryo have been achieved [45,46]. Several studies demonstrated that the oviductal fluid plays an important biological role in sperm capacitation [47], fertilization [48], and embryo development [49]. Recently, EVs derived from oviductal fluids, called oviductosomes, have been emerged as key factors of the early embryo-maternal communication [50,51]. 

It has been demonstrated that bovine oviduct-derived EVs improved embryo quality through increasing the number of trophectoderm and total blastocyst cells and survival after embryos vitrification/warming [52]. Additionally, EVs derived from isthmus oviductal fluid increase the survival rate and the quality of in vitro produced blastocysts [53]. Furthermore, supplementing EVs from donor oviduct fluid to embryo transfer medium increases birth rate in mice by the downregulation of apoptosis and enhanced cellular differentiation in embryos [54]. Taken together, it can be assumed that EVs derived from the oviduct are closely associated with embryo–maternal interactions during early embryonic development [55]. The oviduct fluid-derived EVs, containing microvesicles and exosomes, were able to be internalized by embryos and increased blastocyst formation rate and extended embryo survival time [42]. Moreover, the EVs derived from oviduct epithelial cells showed a positive effect on the quality of in vitro produced embryos by regulating gene expression in cows [52,53]. 

Additionally, it is noted that early preimplantation embryos can autonomously regulate their own development by producing numerous growth factors, receptors [56], and EVs-enriched components [57]. Recent studies demonstrated that in vitro cultured bovine embryos of days 7–9 secreted higher concentration of EVs compared to those secreted by the embryo from parthenogenesis [58]. Another study suggested the exosomes secreted by cloned bovine embryos improved the viability of cloned embryos and increased implantation rates as well as full-term rate [59]. In line with these observations, we proved that exosomes/microvesicles derived from cloned embryos could be internalized by the embryos and they contained pluripotency mRNA transcripts such as *Oct4*, *Sox2*, *Klf4*, *c-Myc*, and *Nanog* [60]. In this study, it can be assumed that in vitro-produced embryos can secrete exosomes/microvesicles as a potential tool for embryo-to-embryo communication within the microenvironment (Figure 1). In addition, recent studies demonstrated that exposure to oviduct-derived EVs regulated the phospholipid composition of embryos toward phosphatidylcholines, phosphatidylethanolamines, and sphingomyelins [61]. Thus, it can be indicated that the EVs derived from reproductive tract or the embryo itself would be closely associated with the process of embryo development as well as oocyte developmental competence.

The presence of EVs has been identified not only in follicular fluid, oviduct fluid, and embryo culture medium, but also in the uterine lumen. This fluid carries numerous bioactive substances that are essential for the embryo development and pregnancy [62,63,64]. As there is an intense communication between the embryo and maternal uterine environment during early embryo development, it is necessary to understand the role of EVs between embryo–maternal crosstalk. Burns et al. [65] demonstrated that during early pregnancy period in sheep, conceptus-trophectoderm-cell-derived EVs were able to be uptaken by the endometrium of cyclic sheep after labeling with PKH67 fluorescent dye. Uterine fluid-derived EVs in the pre-implantation period upregulate the mRNA transcript levels of apoptotic genes including *BAX*, *CASP3*, *TP53*, and *TNFA* in the endometrium cells [66]. Additionally, exosomes derived from uterine fluid in pregnant cow were treated to trophoblast cells, and *IFNT* and *CDX2* mRNA transcript levels did not change [66] while they induced IFNT-associated genes [67], indicating that the components of EVs might be closely involved in the process of pregnancy. These findings suggest that the EVs mediate the paracrine communication between the endometrium and conceptus and, consequently, contribute to the maintenance of pregnancy.

## 3. Unique Reproductive Characteristics in Canine Oocyte Maturation

### 3.1. Reproductive Physiology of the Female Dogs

Reproductive physiology in female dogs shows unique characteristics and a clear difference with other mammals such as cats, pigs, cows, and horses. Domestic bitches show non-seasonal monoestrus cycle, i.e., ovulate only once or twice per year. The canine estrous cycle is classified into four recurring stages: proestrus, estrus, diestrus, and anestrus [82]. The unique reproductive characteristic in bitches is that ovarian follicles release immature prophase I oocytes that require an additional 48–72 h to undergo maturation in the oviductal canal [83,84]. It is wellknown that the dog is the only domestic animal in which luteinization of the preovulatory follicles occurs with increased progesterone level before the ovulation [85]. Therefore, the preovulatory rise in progesterone coincides with the preceding estrogen peak in eliciting estrus behavior in bitches [85]. In addition, the cytoplasm of canine oocytes appears relatively dense and darker than those of other mammals because of a high amount of lipid droplets [86].

### 3.2. Physiological Characteristic of Oviduct

The mammalian oviduct is the place where both male and female gametes are transported and stored for maturation and finally fertilized. Specifically, the canine oviduct is divided into total three parts: infundibulum, ampulla, and isthmus [87,88]. The infundibulum is the region where there is a funnel-shaped dilation covering the ovulation fossa that transports the immature oocyte to the ampullary tract. Fertilization occurs in the ampulla, and early stage embryos travel to the isthmus region of the oviduct and enter the uterus at the morula or early blastocyst stage [89]. Unlike other mammals, dog oocytes reside in the oviduct for an extended period as they require additional 48–72 h to complete maturation within the ampulla region of the oviduct. Therefore, immature canine oocytes at the germinal vesicle stage survive longer than other mammals’ oocytes in the oviductal tract [90,91,92]. It has been shown that canine oocytes sustain the viability and competency for up to seven days after ovulation in the oviduct [93]. Therefore, the oviductal microenvironment is pivotal for the maturation process and prolonged period of survival of oocytes, and it is considered as one of the most peculiar and distinguished reproductive characteristics in the dog.

Consequently, several reports have been focused on the reciprocal communication between the oocytes and the oviductal epithelium in vivo [94,95]. Moreover, it has been demonstrated that co-culture oviduct epithelial cells with canine oocytes positively influenced the oocyte in vitro maturation (IVM) [96,97]. Additionally, superoxide dismutase activity, a primary antioxidant enzyme that restricts reactive oxygen species, showed significant increases in oviductal fluid in estrus bitches when compared to those in anestrus and diestrus bitches [98]. Therefore, it can be assumed that oviduct cells and their derivatives would involve in canine in vivo oocyte maturation, and further studies are required to discover the specific oviductal factors involving in this process.

### 3.3. In Vitro Maturation of Canine Oocytes

In a large number of domestic mammals, assisted reproductive techniques (ARTs) are wellestablished, including gamete manipulation, in vitro oocyte maturation, and preimplantation embryo development. However, there are comparative limitations and obstacles for developing ARTs in canid species because of their unique and peculiar reproductive characteristics [92,99,100,101]. To date, numerous studies identified some of the factors that mediate oocyte development and maturation in vitro; however, still, there is a much lower rate of in vitro maturation in the dog compared with most of the other domestic mammals [97,100,102,103]. The in vitro maturation rate of canine oocytes is still approximately ~20% and influenced by various experimental conditions such as donor age [83], supplementation of hormones [104], the use of synthetic oviductal fluid [105], oviductal cell co-culture [97,105,106], and the stages of the reproductive cycle [107]. Therefore, new approaches are needed to improve the understanding the mechanisms regulating canine oocyte maturation in vitro. Specifically, studies aiming at assessing the roles of oviductal secretomes in canine oocyte development would establish a new paradigm that is critical for the development of effective canine in vitro maturation system.

## 4. The Physiological Function of Oviduct-Derived Exosomes on Cumulus-Oocyte Complexes in Dogs

The oviduct provides biochemical and biophysical support for gametes maturation, fertilization, and embryo development [96,108,109]. The reproductive function of ovarian follicle and uterus has been well established; however, the role of oviduct in reproduction system is comparably underestimated. The oviduct epithelial cells secrete proteins that form oviductal fluid [43,110]. Oviductal fluid is composed of various components such as carbohydrate, proteins, and lipids [46]. Transcriptomics analysis showed that oviduct fluid compositions are different depending on the estrus cycle stage in human [111,112], cow [113,114], and pig [115]. Moreover, proteomics analysis reported that a similar protein composition is shown during the follicular and luteal phase of cycle in pig [116], while there are significant differences in oviductal fluid proteome in cow and sheep during estrus cycle [117,118]. Likewise, it would be valuable study to identify EVs proteome in oviductal fluid and evaluate their biological functions considering the temporal and spatial comparisons. Recently, a total of 319 proteins from in vivo and in vitro oviductal EVs were identified, and 97 were exclusively expressed in in vivo, while 46 were differentially expressed only in in vitro in cow [42]. Analysis showed that the main functions of these identified protein were related to sperm–oocyte binding, fertilization, and embryo development [42]. Another study indicated that EVs derived from bovine oviduct cells improved blastocyst quality and showed cryoprotective effects after in vitro culture [52].

Up until now, the properties of EVs have not been thoroughly investigated in canine reproduction system, particularly in the aspect of oocyte maturation. As the immature oocyte requires most of the biological sources for energy metabolism from the surrounding somatic cells [119,120], the communication between oocytes and their surrounding somatic cells is necessary to transport various regulatory molecules [121] for oocyte development. Recently, Lange-Consiglio et al. [80] have demonstrated that the oviduct-derived microvesicles can be abundantly incorporated by cumulus cells, but not in the oocyte cytoplasm after 48 h of culture, and the microvesicles within the cytoplasm can be observed after 72 h of culture. Moreover, Lee et al. [81] showed that canine oviduct-derived exosomes can be preferentially taken up by the cumulus cells during 24–48 h culture, and the oviductal exosomes are clearly visible within the cytoplasm after 72 h culture. In other mammals, it has been suggested that EVs from ovarian follicles were preferentially taken up by granulosa cells in cow [122], and oviductal fluid-derived EVs interacted with the cumulus cells, zona pellucida, and oocyte, being able to cross the zona pellucida in pig [73]. Therefore, it can be assumed that EVs would act as messengers for cell-to-cell communication for oocyte development.

A recent study showed the involvement of exosome-like vesicles in small ovarian follicles stimulated cumulus expansion and upregulated cumulus gene expression (*PTGS2*, *PTX3*, and *TNFAIP6*) in vitro in bovine species [39]. In horses, follicular exosome-like vesicles regulate the expression levels of transcripts related to transforming growth factor-β and bone morphogenetic protein signaling in mural granulosa cells [123,124]. Furthermore, it has been demonstrated that exosomal miRNAs from human follicular fluid are involved in molecular pathways regulating follicle growth [36]. Among these exosomal miRNAs, nine of them mediate mRNAs expression encoding inhibitors of follicle maturation and meiotic resumption. In contrast, exosome-like vesicles derived from porcine follicular fluid exerted no effect on cumulus expansion and the expression levels of transcripts required for the normal expansion process [125]. For the dog, it has been demonstrated that oviductal exosomes effectively regulate overall physiological condition of cumulus cells such as cell replication, viability, accumulation of reactive oxygen species, and the apoptotic rate [79]. Moreover, it has been suggested that oviduct-derived exosomes would be the primary mediators of molecular interaction among cumulus cells via EGFR (epidermal growth factor receptor)/MAPK (mitogen-activated protein kinase) signaling pathway [81]. Collectively, the potential physiological functions of EVs on follicular somatic cells are shedding light on the improvement of oocyte and embryo development in vitro.

It has been demonstrated that EVs of oviduct epithelium effectively induced favorable biochemical milieu able to improve in vitro oocyte maturation [94,96,97,105,126]. A recent study showed that canine microvesicles contain miR-30b, miR 375, and miR 503 [80], and these miRNAs are involved in various signaling pathways such as WNT, MAPK, ERbβ, and transforming growth factor beta (TGFβ) which are closely related to follicular growth and oocyte maturation [27,33,36]. In contrast, these miRNAs were found in follicular fluid where the oocyte maturation occurs in the cow [33]. These findings are considered particularly interesting for the canine reproduction system because various miRNAs in EVs showed potential roles for gamete maturation, fertilization, and embryo development from the previous literature [34,127,128,129,130,131]. Therefore, it can be assumed that these miRNAs are present in the oviduct cells as canine oocyte maturation occurs in the oviductal canal. 

Particularly, TGFβ and MAPK are one of the key molecular pathways to mediate the meiotic resumption. The growth differentiation factor 9 (GDF9) and bone morphogenic protein 15 (BMP15), which belong to the members of the TGFβ superfamily, are correlated with the oocyte growth from very early stages and are responsible for follicular growth and oocyte maturation in the human [132], mouse [133,134], dog [81,107,135], pig [136], cow [137], and fish [138]. Recent study demonstrated that oviduct-derived exosomes significantly increased the protein levels of GDF9 and BMP15 in cytoplasm of canine oocytes during IVM [81]. Interestingly, the EVs derived from bovine follicular fluid regulate the GDF9/BMP15 transcript levels to a lesser extent in thermoneutral conditions of oocytes compared to heat stress condition of oocytes [70]. This finding indicates that EVs might actively involve in oocyte development when they are exposed into suboptimal environment.

The activation of MAPK signaling pathway promotes oocyte meiotic resumption, germinal vesicle breakdown, and spindle microtubule organization [139,140,141]. During in vitro maturation of oocyte, gradually increasing level of MAPK phosphorylation has been demonstrated in murine [142], bovine [143], ovine [144], and canine [81] oocytes. A recent study suggested that EVs derived from feline follicular fluid contain proteins that are involved in MAPK and phosphatidylinositol 3-kinase (PI3K) signaling pathway, and these vesicles enhanced the ability of frozen/thawed oocytes to resume meiotic resumption [78]. Additionally, it has been found that the upregulated miRNAs in EVs derived from follicular fluid are important tools for modulating the MAPK pathway in the cow [145]. In the horse, bioinformatic analysis revealed that miRNAs present in EVs of follicular fluid could target MAPK signaling and focal adhesion as the most prominent pathways [27]. In addition, human follicular fluid-derived EVs contain miRNAs which are associated with growth, development, and signaling pathways including MAPK [35]. In line with their results, canine oviduct-derived exosomes exerted positive effects by upregulating the gene/protein expression levels related with the EGFR/MAPK signaling pathway [81]. Moreover, we demonstrated that the co-incubation of canine exosomes with oocytes markedly enhanced the in vitro maturation rate up to 2.5 times higher than those matured without exosomes. As previous studies implicated the cues of oocyte itself as well as the activation of EGFR/MAPK signaling in oocytes are required for successful meiosis [146,147], it can be assumed that canine oviduct-derived exosomes would play a necessary role through transporting essential genetic materials to oocytes. 

Consequently, recent numerous articles clearly showed that extracellular vesicles derived from reproductive tract play a crucial role in oocyte maturation, which is a prerequisite process for reproductive success. In particular, the key role of exosomes on signaling pathways relevant to oocyte, embryo, and fetus development has been actively discovered and summarized in Figure 2. Although still further systematic and specific studies regarding exosomes with canine oocytes are required to establish in vitro maturation system, the research in the field of reproductive EVs in dogs has been gradually increasing, and would provide better understanding of the function of canine EVs on oocyte maturation.

## 5. Concluding Remarks

In the last decade, the physiological functions of exosomes have been studied exponentially, while the physiological relationship between exosomes and reproductive aspects is not entirely understood. In line with recent research for exosomes on reproduction, this review discussed how exosomes play a crucial role in reproductive processes, especially during oocyte development in mammals, including dogs. Therefore, this article brings new insights into the contribution of exosomes as potential candidates for regulating oocyte maturation in canine reproduction, which would suggest a new avenue for establishing novel in vitro maturation, in vitro fertilization, and embryo in vitro culture systems. In addition, it is evident that the application of exosomes in ARTs in canine reproduction will be highly warranted to broaden our knowledge of other mammals’ reproduction as well as human research in the near future.

## Figures and Tables

**Figure 1 animals-10-01971-f001:**
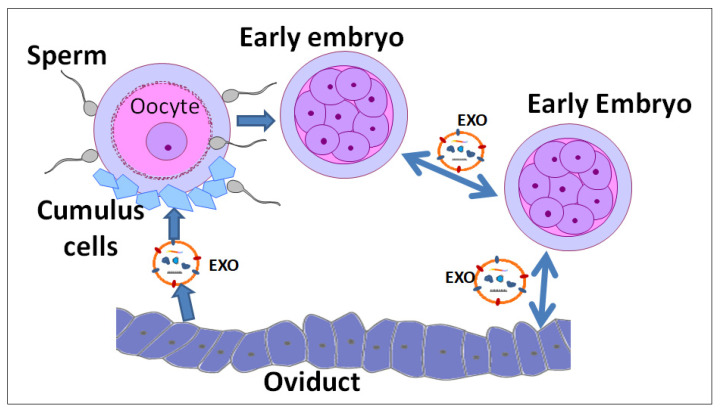
The proposed interaction between the oviduct, cumulus-oocyte complex, sperms, and embryos through the exosomes (EXO). Exosomes contain bioactive cargo molecules such as mRNA, miRNAs, and proteins; they are transferred from secreting cells and regulate the physiological processes in the target cells.

**Figure 2 animals-10-01971-f002:**
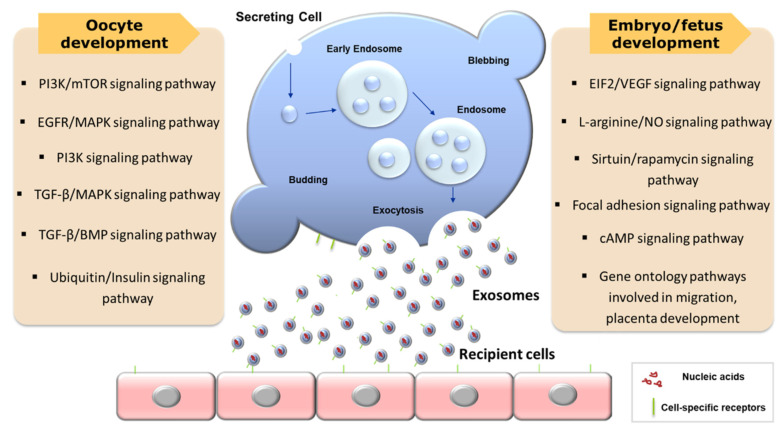
The representative role of exosomes on signaling pathways for oocyte and embryo/fetus development.

**Table 1 animals-10-01971-t001:** Recent studies related with the effect of extracellular vesicles on gametes/ovary/cumulus cells/embryos development.

Species	Types of Extracellular Vesicles	Origin of Extracellular Vesicles	Finding	Year	Reference
Human	Extracellular vesicles	Follicular fluid	Extracellular vesicles’ miRNAs play a role in pathways of ovarian function and follicle development.	2018	[35]
Human	Extracellular vesicles	Preimplantation embryo	Preimplantation embryos at all developmental stages secrete extracellular vesicles.	2019	[68]
Cow	Extracellular vesicles	Oviduct cells	In vivo extracellular vesicles contain proteins involved in sperm–oocyte binding, fertilization, and embryo development.	2017	[42]
Cow	Exosomes	Follicular fluid	Exosomes in follicular fluid play important roles during oocyte maturation and protect the gamete against heat stress.	2019	[38]
Cow	Extracellular vesicles	Oviduct cells	Extracellular vesicles from the isthmus improve the development and quality of embryos.	2017	[53]
Cow	Exosomes	Granulosa cells	Exosomes alter the cellular oxidative stress response molecules, including Nrf2, CAT, PRDX1, and TXN1 in granulosa cells.	2017	[69]
Cow	Extracellular vesicles	Follicular fluid	Extracellular vesicles show a lesser expression of genes related to oocyte quality such as *GDF9*, *BMP15*, *IGFBP2*, *CDCA8*, and *STAT3*, when the oocytes are exposed to the thermoneutral condition compared to the heat stress condition.	2019	[70]
Cow	Extracellular vesicles	Follicular fluid	Extracellular vesicles from follicular fluid modulate the arrest of oocyte meiosis, similar to the C-type natriuretic peptide–natriuretic peptide receptors subtype 2 system.	2020	[71]
Cow	Extracellular vesicles	Oviduct fluid	Extracellular vesicles from oviduct fluid are taken up by the embryo; they alter lipid composition and modulate lipid metabolism.	2020	[61]
Pig	Exosomes	Follicular fluid	Exosomal mRNAs are enriched in encoding proteins involved in metabolic, PI3K-AKT, and MAPK pathways.	2019	[72]
Pig	Extracellular vesicles	Oviduct	Extracellular vesicles are able to participate in maintaining sperm viability and reducing motility, functions associated with the oviduct sperm reservoir.	2020	[73]
Mouse	Extracellular vesicles	Oviduct	Extracellular vesicles carry and deliver tyrosine phosphorylated proteins to sperm and mediate an upregulation of plasma membrane Ca^2+^ ATPase (PMCA) 1 in *Pmca4*^−/−^ female mice during proestrus/estrus.	2018	[74]
Mouse	Extracellular vesicles	Vaginal luminal fluid	Extracellular vesicles deliver proteins involved in preventing premature capacitation and acrosome reaction.	2019	[75]
Mouse	Extracellular vesicles	Mesenchymal stem cells	Stem cell-derived extracellular vesicles increase the quality of the embryo by modulating antioxidant and pluripotent genes.	2019	[76]
Cat	Extracellular vesicles	Oviduct cells	Extracellular vesicles prevent premature acrosome exocytosis in cheetah sperm, and feline extracellular vesicles carry proteins that have the potential to restore sperm function after cryopreservation.	2020	[77]
Cat	Extracellular vesicles	Follicular fluid	Extracellular vesicles enhance the ability of frozen/thawed oocytes to resume meiotic resumption, and proteomic analysis of extracellular vesicles identified proteins that are involved in oocyte meiosis, oxidative phosphorylation, and MAPK and PI3K-AKT signaling pathways	2020	[78]
Dog	Exosomes	Oviduct cells	Exosomes enhance the physiological condition of cumulus cells via EGFR/MAPK signaling pathway.	2020	[79]
Dog	Microvesicles	Oviduct cells	Microvesicles are involved in cellular trafficking and exert a positive effect during oocyte maturation.	2017	[80]
Dog	Extracellular vesicles	Oviduct cells	Thawing red wolf sperm in the presence of canine extracellular vesicles improves post-thaw motility and prevents premature acrosome exocytosis and canine extracellular vesicles carry proteins that have the potential to restore sperm function after cryopreservation.	2020	[77]
Dog	Exosomes	Oviduct cells	Exosomes markedly enhance canine oocyte maturation and modulate gene/protein levels relevant to the EGFR/MAPK pathway.	2020	[81]

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
