# Peer review of "Exosomes as a Potential Tool for Supporting Canine Oocyte Development"

_animals, 2020, doi:10.3390/ani10111971_

Round 1

Reviewer 1 Report

Thank you for the reply to the comments.

Author Response

Thank you for your thoughtful consideration about our manuscript.

Reviewer 2 Report

The present review discusses the role of extracellular vesicles (EV) on the gamete and embryo development. The authors focus pretty much on the exosomes type of EV. In the first part of the work, the authors summarize the literature on EVs and their effect on oocyte growth and embryo development. Then, the authors focus on the EVs in dog. Overall, the review is nice and could be of interest for the scientific community related to this field. English needs moderate editing in my opinion. However, the first part of the work is too generic. In my opinion, It is missing a part that describe the content of EVs. A big part of the literature on EVs discusses indeed the content of the exosomes, which is pretty much represented by microRNAs. Therefore, the authors are fully recommended to include a paragraph that describes this topic before going through their discussion on the findings in the dog. They also talk about microRNAs at some point and they do not explain what they are and their role on the oogenesis and embryo development. In conclusion, the manuscript could be taken into account for publication after they have included this part.

Some recent references recommended to discuss this part:

Pasquariello R, Manzoni EFM, Fiandanese N, Viglino A, Pocar P, Brevini TAL, Williams JL, Gandolfi F. Implications of miRNA expression pattern in bovine oocytes and follicular fluids for developmental competence. Theriogenology. 2020 Mar 15;145:77-85. doi: 10.1016/j.theriogenology.2020.01.027. Epub 2020 Jan 20. PMID: 32004821.

Pasquariello R, Fernandez-Fuertes B, Strozzi F, Pizzi F, Mazza R, Lonergan P, Gandolfi F, Williams JL. Profiling bovine blastocyst microRNAs using deep sequencing. Reprod Fertil Dev. 2017 Aug;29(8):1545-1555. doi: 10.1071/RD16110. PMID: 27623773.

Gad A, Nemcova L, Murin M, Kanka J, Laurincik J, Benc M, Pendovski L, Prochazka R. microRNA expression profile in porcine oocytes with different developmental competence derived from large or small follicles. Mol Reprod Dev. 2019 Apr;86(4):426-439. doi: 10.1002/mrd.23121. Epub 2019 Feb 13. PMID: 30756429.

Sohel MM, Hoelker M, Noferesti SS, Salilew-Wondim D, Tholen E, Looft C, Rings F, Uddin MJ, Spencer TE, Schellander K, Tesfaye D. Exosomal and Non-Exosomal Transport of Extra-Cellular microRNAs in Follicular Fluid: Implications for Bovine Oocyte Developmental Competence. PLoS One. 2013 Nov 4;8(11):e78505. doi: 10.1371/journal.pone.0078505. PMID: 24223816; PMCID: PMC3817212.

Andronico F, Battaglia R, Ragusa M, Barbagallo D, Purrello M, Di Pietro C. Extracellular Vesicles in Human Oogenesis and Implantation. Int J Mol Sci. 2019 May 1;20(9):2162. doi: 10.3390/ijms20092162. PMID: 31052401; PMCID: PMC6539954.

Author Response

Reference No.: animals-962208

Responses to reviewer 2

Thank you for the comments. All the details pointed out greatly contributed to improving our manuscript. We have substantially revised the manuscript and provided the list of changes as follows.

Specific comments

Reviewer: 2

Q1) The present review discusses the role of extracellular vesicles (EV) on the gamete and embryo development. The authors focus pretty much on the exosomes type of EV. In the first part of the work, the authors summarize the literature on EVs and their effect on oocyte growth and embryo development. Then, the authors focus on the EVs in dog. Overall, the review is nice and could be of interest for the scientific community related to this field. English needs moderate editing in my opinion. However, the first part of the work is too generic. In my opinion, It is missing a part that describe the content of EVs. A big part of the literature on EVs discusses indeed the content of the exosomes, which is pretty much represented by microRNAs. Therefore, the authors are fully recommended to include a paragraph that describes this topic before going through their discussion on the findings in the dog. They also talk about microRNAs at some point and they do not explain what they are and their role on the oogenesis and embryo development. In conclusion, the manuscript could be taken into account for publication after they have included this part.

A1) Thank you for your suggestion. First, as reviewer suggested, we’ve taken the English editing service (Grammaly premium service) to revise the grammatical errors. Also, Islam M Saadeldin who is a new co-author and an expert in the EVs fields, has performed English editing in details.

In addition, we have added the information of the content of the exosomes, and explain what microRNAs are and their role on the oogenesis and embryo development in line 48-65 and line 126-138, respectively, in revised highlight manuscript as follow:

Line 48-65: A typical EVs contains protein, lipids, and genetic materials enclosed by a lipid bilayer membrane that originated from the plasma membrane [5]. The EVs content are determined by several factors such as the type, physiological and pathological status of the donor cells [6]. Proteins are an important part of EVs since the inner and outer domains of proteins determine the functional capacity of EVs [7]. Also, some proteins determine the EVs origin and target, including antigen presentation proteins, immunoglobulins, cell surface proteins, and cell-to-cell specific signaling proteins [8,9]. Apart from proteins, a great quantity of lipids is found in EVs, however, still, their composition has not been studied compared with the protein constituent. EVs contain high contents of cholesterol, ceramide, or other sphingolipids, and phosphoglycerides with long and saturated fatty-acyl chains [10]. The lipid content contributes to the biogenesis of the EVs by transporting hormone-like molecules and regulating homeostasis in the recipient cells [11]. In addition, it has been demonstrated that EVs contain a large amount of RNAs such as mRNA, miRNA, and long non-coding RNA [12]. In particular, miRNA plays an important role in regulating gene expression to recipient cells [13]. Moreover, a recent study proved that double-stranded DNA has been found in the EVs derived from the cancer cells, and the release of DNA also maintains the homeostasis in recipient cells [14]. Also, the polysaccharides and glycans are located on the outer surface of EVs, as the least components of EVs. These factors are composed of mannose, α-2-3, and α-2-6 sialic acids, complex-n-linked glycans, and polylactosamine [15].

Line 126-138: Specifically, miRNAs are critically involved in RNA silencing and post-transcriptional regulation of gene expression in the cells [43,44]. It has been demonstrated that the changes in microRNA expression were positively correlated with progesterone synthesis in bovine follicular fluids however it was associated with lower mitochondrial function in the oocytes [45]. Moreover, miRNA could regulate the maternal mRNA storage and synthesis in the bovine ooplasm [45]. Interestingly, specific miRNAs were found to be essential of follicular growth and oocyte developmental competence; miR-769, miR-1343, miR-450a, miR-204, miR-1271 and miR-451 were found in bovine follicular fluid and regulate follicular growth and acquisition of oocyte developmental competence [45]. Likewise, deep RNA sequencing revealed several miRNAs involved in follicular growth and oocyte developmental competence in porcine such as MiR-205, miR-16, miR-148a-3p, and miR-125b [46] and in bovine such as miR-654-5p, miR-640, miR-19b-1 and miR-29c [47]. In human, miR-214, miR-454 and miR-888 differential expression were associated with good quality embryo and successful IVF cycles [48-50].

Q2) Some recent references recommended to discuss this part:

1) Pasquariello R, Manzoni EFM, Fiandanese N, Viglino A, Pocar P, Brevini TAL, Williams JL, Gandolfi F. Implications of miRNA expression pattern in bovine oocytes and follicular fluids for developmental competence. Theriogenology. 2020 Mar 15;145:77-85. doi: 10.1016/j.theriogenology.2020.01.027. Epub 2020 Jan 20. PMID: 32004821.

2) Pasquariello R, Fernandez-Fuertes B, Strozzi F, Pizzi F, Mazza R, Lonergan P, Gandolfi F, Williams JL. Profiling bovine blastocyst microRNAs using deep sequencing. Reprod Fertil Dev. 2017 Aug;29(8):1545-1555. doi: 10.1071/RD16110. PMID: 27623773.

3) Gad A, Nemcova L, Murin M, Kanka J, Laurincik J, Benc M, Pendovski L, Prochazka R. microRNA expression profile in porcine oocytes with different developmental competence derived from large or small follicles. Mol Reprod Dev. 2019 Apr;86(4):426-439. doi: 10.1002/mrd.23121. Epub 2019 Feb 13. PMID: 30756429.

4) Sohel MM, Hoelker M, Noferesti SS, Salilew-Wondim D, Tholen E, Looft C, Rings F, Uddin MJ, Spencer TE, Schellander K, Tesfaye D. Exosomal and Non-Exosomal Transport of Extra-Cellular microRNAs in Follicular Fluid: Implications for Bovine Oocyte Developmental Competence. PLoS One. 2013 Nov 4;8(11):e78505. doi: 10.1371/journal.pone.0078505. PMID: 24223816; PMCID: PMC3817212.

5) Andronico F, Battaglia R, Ragusa M, Barbagallo D, Purrello M, Di Pietro C. Extracellular Vesicles in Human Oogenesis and Implantation. Int J Mol Sci. 2019 May 1;20(9):2162. doi: 10.3390/ijms20092162. PMID: 31052401; PMCID: PMC6539954.

A2) We thank the reviewer for this helpful suggestion, we cited some of the suggested references in the first version, and we profoundly demonstrated the roles of miRNA in the revised text taken in consideration all of these important references. In addition, we have decided not to cite the #2 reference because it is irrelevant to the context as it describes the blastocyst miRNAs derived from the blastocysts itself, however, we focus on the effects on the oocyte developmental competence.

Reviewer 3 Report

AA revised the ms according to the reviewers' comments. The paper is now suitable for pubblication

Author Response

(The authors gave the same response as above.)

Round 2

Reviewer 2 Report

I thank you the authors to fully address my comments. Therefore, the manuscript can be accepted for publication at this time.

One minor comment is related to the use of acronyms such microRNAs, which has been used as miRNAs or microRNAs. I recommend the use of microRNAs for the first time in the text and then the contract short word miRNAs.

Author Response

Reference No.: animals-962208

Responses to reviewer 2 

Thank you for the comments. All the details pointed out greatly contributed to improving our manuscript. We have substantially revised the manuscript and provided the list of changes as follows. 

Specific comments

Reviewer: 2

Q1) One minor comment is related to the use of acronyms such microRNAs, which has been used as miRNAs or microRNAs. I recommend the use of microRNAs for the first time in the text and then the contract short word miRNAs.

A1) Thank you for your suggestion. We have used microRNAs for the first time in line 54 in revised highlight manuscript and then the contract short word miRNAs.

This manuscript is a resubmission of an earlier submission. The following is a list of the peer review reports and author responses from that submission.

Round 1

Reviewer 1 Report

This is a very thorough and informative review on the topics of canine oocyte maturation and the possible influence of extracellular vesicles produced by the oviduct.

Minor comment: the method to isolate oviductal extracellular vesicles mentions centrifugation speeds (10,000 x g) that are lower than in other species (usually 20,000 up to 100,000).  Can authors comment about that?

Author Response

Reference No.: animals-898320

Responses to reviewer 1

Thank you for the comments. All the details pointed out greatly contributed to improving our manuscript. We have revised the manuscript and provided the list of changes as follows.

Specific comments

Minor point

Reviewer: 1

Q1) The method to isolate oviductal extracellular vesicles mentions centrifugation speeds (10,000 x g) that are lower than in other species (usually 20,000 up to 100,000). Can authors comment about that?

A1) Thank you for providing these insights. To isolate oviductal exosomes in canine species, we’ve used total exosomes isolation reagent. According to the procedure, it suggests 10,000 x g for 1 hour to collect the exosomes. Also, this centrifugation speed has been applied into other species in recent published articles. The centrifugation speed (20,000 up to 100,000 x g) that reviewer mentioned is usually applied in traditional method to isolate the extracellular vesicles. Generally, the extracellular vesicles can be also obtained by several steps of centrifugation (~100,000 x g) without using commercial reagent.

Reviewer 2 Report

The topic contained within this review is of high interest.  Very little work is often completed on off-model species, and much can be learned from in-depth analysis of the literature on more unique models. 

In its current state, this review appears to be much too highly focused on recent works from the author, and should be expanded to include a more balanced approach. 

Section 4 reads like a methods section, which seems out of place for a review paper, and much of the manuscript concentrates almost exclusively on the author's own data. 

Additionally, there are some errors in referencing which could be concerning; most importantly reference 90 which although stated within the text as one of the only recent publications concerning EVs in canine oocytes, is actually a manuscript published in plant physiology discussing plant immunity. This reviewer will admit that they did not check the rest of the references, but as this is a highly pertinent reference, this is cause for concern. 

In order to be resubmitted, the review would also require extensive english editing.

Author Response

Reference No.: animals-898320

Responses to reviewer 2

Thank you for the comments. All the details pointed out greatly contributed to improving our manuscript. We have substantially revised the manuscript and provided the list of changes as follows.

Specific comments

Reviewer: 2

Q1) In its current state, this review appears to be much too highly focused on recent works from the author, and should be expanded to include a more balanced approach.

A1) Thank you for your suggestion. We’ve totally agreed that the previous paper appeared to be too highly focused on our recent works. Therefore, we have expanded the range of article to be balanced approach. The substantial modification has been applied from section 4 to section 6 as follows:

1) Line 134-190 in revised highlight manuscript:

4.1. Collection of canine oviduct cells and their extracellular vesicles

            Generally, oviduct flushing and scraping method are widely used to isolate canine oviduct cells [11,50,64-66]. Briefly, an inverted needle was inserted into the oviduct bursa from the bursa split and an appropriate volume of flushing medium was injected into the oviduct to its end using a needle or intravenous catheter. Then, centrifugation was performed to collect the oviduct cells followed by washing with PBS to remove blood and tissue debris. For scraping method, the entire oviduct was opened longitudinally and the epithelium was scraped with a microblade into tissue culture medium. After several washes in medium, oviductal epithelial cells were obtained.

            Up to date, only a few researches have been performed to isolate extracellular vesicles derived from canine oviduct cells. It has been suggested that the microvesicles derived from canine oviduct cells can be successfully isolated from conditioned medium [11]. In brief, the multicellular spheroids from oviductal canal were cultured for 3 days and the supernatant was ultracentrifuged at 100,000 x g for 1 h. Then, the pellet was washed and submitted to another ultracentrifugation under the same conditions. Finally, canine microvesicles were obtained. Other recent researches demonstrated that canine extracellular vesicles and exosomes can be easily collected by using a commercial isolation kit [36,67]. Briefly, conditioned medium from oviduct cell culture or flushing medium from oviduct was centrifuged at 2,000 x g for 30 min to remove cells and debris. Then, the isolation reagent was mixed with medium and incubated overnight at 4 °C. Consequently, the oviduct-derived extracellular vesicles pellets can be obtained with centrifugation at 10,000 x g for 1 h.

4.2. Characterization and identification of canine oviduct-derived exosomes

            In general, exosomes are formed by the inward budding of endosomal membranes of cells and released into the extracellular microenvironment by fusion with the recipient cell membrane. According to many other previous researches, the characterization and identification of exosomes have been well-established [12,68-70]. There are total three main characterization methods for the identification of oviduct-derived extracellular vesicles in dogs as previously described [11,35,67].

            First, oviduct-derived exosomes including exosomes can be characterized by using transmission electron microscopy (TEM). Generally, oviduct-derived exosomes were exhibited a circular shaped morphology with appropriate size using TEM analysis. Second, nanoparticle tracking analysis (NTA) can be applied to characterize oviduct-derived exosomes regarding concentration, intensity, and size. Briefly, purified oviduct-derived exosomes were diluted in PBS and applied for recording videos. The concentration of exosomes was adjusted to capture approximately 50 vesicles in the field of view for quantification. The exosomes quantification including mean, mode, and standard division is automatically measured with this analysis. Lastly, oviduct-derived exosomes can be characterized using exosomal specific markers. It has been suggested that exosomes have specific proteins that depend on the cell type as well as a specific subset of cellular proteins [71]. For example, there are three tetraspanin membrane proteins (CD9, CD63, and CD81) classically used as exosomes specific markers [72,73]. In addition, heat shock proteins (HSP)70 and HSP90 are molecular chaperones, and tumor susceptibility gene 101 protein can be used as specific exosomal markers as these proteins are involved in multivesicular bodies biogenesis [74]. Likewise, diverse exosome specific markers have been applied to characterize the exosomes derived from various cell types including oviduct cells.

2) Line 249-270 in revised highlight manuscript:

            There are several critical processes necessary to activate and mature the oocytes within preovulatory follicles. Cumulus cells receive biological signals from the follicular cells including granulosa and theca cells in vivo environment. Cumulus expansion occurs during this period, and this is a process that requires cumulus cells to secrete hyaluronic acid into the extracellular environment [23,75]. Also, a hyaluronan-enriched extracellular matrix makes oocyte and cumulus cells separated and finally induced cumulus expansion. Likewise, various cellular signals are transmitted to the cumulus cells to induce physiological alteration and those cumulus cells can act as a messenger to oocytes by transporting essential signals to them.

            The immature oocyte requires most of substrates for energy metabolism from the surrounding somatic cells [76,77]. Particularly, cumulus cells communicate with the oocyte through gap junctions that transport of various biological molecules [78]. Therefore, cumulus cells have gained attention because of their vital roles in regulating oocyte maturation [79,80]. A recent study showed the involvement of exosome-like vesicles in small ovarian follicles stimulated cumulus expansion and up-regulated cumulus gene expression (PTGS2, PTX3, TNFAIP6) in vitro in bovine species [81], while exosome-like vesicles present in follicular fluid were not efficient in inducing cumulus expansion in porcine species [82]. In canine species, the oviduct-derived exosomes positively affect on physiological conditions of cumulus cells [35]. They exerted their effects on cumulus cells by increasing cell concentration, viability, and proliferation rate, which indicated canine oviduct cell-derived exosomes also show potential biological functions on recipient cells. In addition, it was demonstrated that canine oviduct-derived exosomes were associated with EGFR (epidermal growth factor receptor)/MAPK (mitogen-activated protein kinase) pathway, which is closely related with cumulus cell expansion and oocyte development [83-87].

3) Line 333-395 in revised highlight manuscript:

The interactions of gametes and embryos with the oviduct cells and their secretions at molecular level are necessary to understand early reproductive mechanisms. As overall reproductive processes such as gamete maturation, fertilization, and embryo development occur within the oviduct, the potential physiological ability of oviduct cells has been studied in the field of reproduction [64,88,89]. It has been proved that oviduct epithelium effectively induces favorable biochemical milieu able to improve in vitro oocyte maturation [48,58,64,65,90]. However, up until now, the properties of EVs have not been thoroughly investigated in canine reproduction system, particularly in the aspect of oocyte maturation, because of their unique reproductive characteristics. As canine oocyte undergoes maturation process within the oviductal canal, investigation of oviduct-derived EVs would be an essential step to overcome the difficulty of the establishment of in vitro maturation system. As far as our knowledge, there are only a few articles which investigate the effect of EVs on canine oocyte maturation; 1) Effect of oviduct-derived microvesicles on canine oocyte maturation [11] , 2) Effect of oviduct-derived exosomes on canine oocyte maturation via EGFR/MAPK signaling pathway [36].

It has been suggested that different conditions of canine IVM system including a monolayer of oviduct epithelial cells, multicellular spheroids derived from oviduct, conditioned medium from oviduct culture, and various concentration of oviduct-derived microvesicle showed dissimilar oocyte maturation rate [11]. Among those groups, specific concentration of oviductal microvesicles showed a high rate of maturation (21.82%) compared with other conditions, which reflects secretomes from canine oviduct has beneficial effects on oocyte development. Also, it was demonstrated that cumulus cells could participate in paracrine mechanisms as a messenger between oviduct and oocytes, which indicates cumulus cells up-take a number of extracellular vesicles before the oocyte uptake [11,36]. Interestingly, it was proved canine oviduct cells and their EVs possess several miRNAs such as miR-30b, miR-375, and miR-503, which induce follicular maturation pathways (WNT, MAPK1/3, TGFβ and ubiquitin), and oocyte maturation [25,26,91]. These findings could be considered particularly interesting for the canine IVM system because various miRNAs in extracellular vesicles showed potential roles for gamete maturation, fertilization, and embryo development from the previous literature [92-97].

In addition, another recent study demonstrated that oviduct-derived exosomes markedly exert their potential effects on canine oocyte maturation [36]. According to the results, the canine oviduct-derived exosomes significantly increased the oocyte maturation rate up to 22.5%. Also, it has been proved canine oviductal exosomes exerted their positive effects by up-regulating the various gene expression levels related to upstream/downstream EGFR/MAPK signaling pathway including EGFR, PKA, TACE/ADAM17, MAPK1/3, MAPK14, PTGS2, and TNFAIP6. Moreover, the strong indicators for oocyte maturation such as p-EGFR, p-MAPK1/3, GDF9 (growth differentiation factor 9), and BMP15 (bone morphogenetic protein 15) were markedly enhanced in COCs by oviduct-derived exosomes in dogs. As previous studies implicated the cues of oocyte itself as well as activation of EGFR/MAPK signaling in COCs are required for successful oocyte maturation [98,99], it can be assumed that canine oviduct-derived exosomes would play a role as a messenger to cumulus cells/oocytes by transporting essential genetic materials to improve COCs development as described in Figure 1.

Recently, it was revealed that canine oviduct-derived extracellular vesicles exclusively expressed 51 proteins from peri-ovulatory and post-ovulatory bitches [67]. Among these proteins, the testis-specific gene antigen 10 was the most abundant protein, which has been known to improve sperm morphology and motility [100]. In addition, they identified various types of proteins within canine oviduct extracellular vesicles including Crisp-3, ATP2B4, and PMCA, which involve in the sperm motility and calcium efflux pumps by maintaining low resting cytosolic calcium gradient across the sperm plasma membrane [101-104]. In line with their results, our recent proteomic data for canine oviduct exosomes also proved their potential physiological functions on not only sperm quality but also oocyte maturation and folliculogenesis (unpublished data). Briefly, the proteome for canine oviduct-derived exosomes indicated significant associations with translation, biosynthetic process, cellular amide metabolic process, and generation of metabolites and energy. Moreover, ingenuity pathway analysis indicated that key pathways essential to oocyte maturation, folliculogenesis, and embryo development were identified in oviduct-derived exosome proteome. In addition to proteins, it has been demonstrated canine oviduct-derived extracellular vesicles also contain lipids that can play a protective role in sperm thawing by modulating plasma membrane conditions [67]. As the composition of lipids in reproductive tract EVs is similar to other EVs which is secreted from other cell types [105], it can be assumed that female reproductive tract-derived EVs have abundant biological information relevant to lipid composition. Although still further systematic and specific research regarding exosomes with canine oocyte is required to establish canine in vitro maturation system, the present studies for canine reproduction with exosomes would bring new insights into canine reproduction system.

Q2) Section 4 reads like a methods section, which seems out of place for a review paper, and much of the manuscript concentrates almost exclusively on the author's own data.

A2) Thank you for providing these insights. As reviewer suggested, we have modified Section 4 substantially. However, as canine extracellular vesicle study has not been well investigated, we hope this small part of method section could be helpful to pioneer who would like to study in this field. Therefore, we have substantially reduced the contents of this section by referring to the recent other researches as follows in line 134-190 in revised highlighted manuscript:

1) Line 134-190 in revised highlight manuscript:

4.1. Collection of canine oviduct cells and their extracellular vesicles

            Generally, oviduct flushing and scraping method are widely used to isolate canine oviduct cells [11,50,64-66]. Briefly, an inverted needle was inserted into the oviduct bursa from the bursa split and an appropriate volume of flushing medium was injected into the oviduct to its end using a needle or intravenous catheter. Then, centrifugation was performed to collect the oviduct cells followed by washing with PBS to remove blood and tissue debris. For scraping method, the entire oviduct was opened longitudinally and the epithelium was scraped with a microblade into tissue culture medium. After several washes in medium, oviductal epithelial cells were obtained.

            Up to date, only a few researches have been performed to isolate extracellular vesicles derived from canine oviduct cells. It has been suggested that the microvesicles derived from canine oviduct cells can be successfully isolated from conditioned medium [11]. In brief, the multicellular spheroids from oviductal canal were cultured for 3 days and the supernatant was ultracentrifuged at 100,000 x g for 1 h. Then, the pellet was washed and submitted to another ultracentrifugation under the same conditions. Finally, canine microvesicles were obtained. Other recent researches demonstrated that canine extracellular vesicles and exosomes can be easily collected by using a commercial isolation kit [36,67]. Briefly, conditioned medium from oviduct cell culture or flushing medium from oviduct was centrifuged at 2,000 x g for 30 min to remove cells and debris. Then, the isolation reagent was mixed with medium and incubated overnight at 4 °C. Consequently, the oviduct-derived extracellular vesicles pellets can be obtained with centrifugation at 10,000 x g for 1 h.

4.2. Characterization and identification of canine oviduct-derived exosomes

            In general, exosomes are formed by the inward budding of endosomal membranes of cells and released into the extracellular microenvironment by fusion with the recipient cell membrane. According to many other previous researches, the characterization and identification of exosomes have been well-established [12,68-70]. There are total three main characterization methods for the identification of oviduct-derived extracellular vesicles in dogs as previously described [11,35,67].

            First, oviduct-derived exosomes including exosomes can be characterized by using transmission electron microscopy (TEM). Generally, oviduct-derived exosomes were exhibited a circular shaped morphology with appropriate size using TEM analysis. Second, nanoparticle tracking analysis (NTA) can be applied to characterize oviduct-derived exosomes regarding concentration, intensity, and size. Briefly, purified oviduct-derived exosomes were diluted in PBS and applied for recording videos. The concentration of exosomes was adjusted to capture approximately 50 vesicles in the field of view for quantification. The exosomes quantification including mean, mode, and standard division is automatically measured with this analysis. Lastly, oviduct-derived exosomes can be characterized using exosomal specific markers. It has been suggested that exosomes have specific proteins that depend on the cell type as well as a specific subset of cellular proteins [71]. For example, there are three tetraspanin membrane proteins (CD9, CD63, and CD81) classically used as exosomes specific markers [72,73]. In addition, heat shock proteins (HSP)70 and HSP90 are molecular chaperones, and tumor susceptibility gene 101 protein can be used as specific exosomal markers as these proteins are involved in multivesicular bodies biogenesis [74]. Likewise, diverse exosome specific markers have been applied to characterize the exosomes derived from various cell types including oviduct cells.

Q3) Additionally, there are some errors in referencing which could be concerning; most importantly reference 90 which although stated within the text as one of the only recent publications concerning EVs in canine oocytes, is actually a manuscript published in plant physiology discussing plant immunity. This reviewer will admit that they did not check the rest of the references, but as this is a highly pertinent reference, this is cause for concern.

A3) Thank you for providing these insights. I have thoroughly checked all references and I put some irrelevant references by mistake. Here is wrong references that I put in original version of manuscript and I’ve modified it as follows:

1) Reference 90 in original paper is changed into Reference 11

2) Reference 43 (Title: Various indices of cellular immunity in cancer and chronic diseases of the stomach) in original paper was deleted.

3) Reference 54 (Title: Influence of hemoglobin saturation on temperature correction of measured blood PO2) in original paper was deleted.

Thank you again for providing these critical point.

Q4) In order to be resubmitted, the review would also require extensive english editing.

A4) Thank you for your suggestion. As you suggested, this review have had English editing.

Round 2

Reviewer 2 Report

Thank you for the revisions to the current manuscript.  Unfortunately there is still major english revisions to be completed prior to publication.  This reviewer suggests enlisting an english editing service prior to resubmission.  Unfortunately the grammatical errors detract from the science presented within this work. 

Many of the edited sections are improved for content, and the review is less focused on the work of the authors, but I still remain uncertain of methodology being presented in the context of a review paper.  Methodological work performed in your own lab should be presented in a research article and referenced as such within a review. A review is not a place to present primary data and methods.